# The Evaluation and Sources of Heavy Metal Anomalies in the Surface Soil of Eastern Tibet

**Mingguo Wang** [1,2] **, Li Yang** [1,*]**, Jingjie Li** [1] **and Qian Liang** [2]

1   Center for Hydrogeology and Environmental Geology Survey, CGS, Baoding 071051, China
2   Henan Academy of Geology, Zhengzhou 450016, China
*   Correspondence: yli_b@mail.cgs.gov.cn

**Abstract:** With the rapid development of the economy, heavy metal soil pollution causes ecosystem deterioration and raises serious concerns. Topsoil samples ($n$ = 205) were collected to investigate the pollution characteristics, risk levels, and pollution sources of heavy metals in the topsoil of eastern Tibet. Heavy metal contents, such as As, Hg, Pb, Cr, Mn, Mo, Ni, Cu, Zn, and Cd, in the soil were tested, and the potential sources were analyzed using correlational and principal component analysis. The results showed high content levels of Cd and Hg, which were 1.42 and 2.45 times higher than the background values of the Tibet Plateau at the beginning of this century, respectively. The enrichment factor (EF), geoaccumulation index (Igeo), and Nemero composite index (PN) showed that Cd and Hg were the main pollutants due to higher traffic flow and mining activities, but the pollution degree was generally not high and was relatively concentrated in the central and northern parts. The results of the principal component analysis showed that the heavy metals in the soil of eastern Tibet were mainly affected by natural factors and traffic factors, and mining activities and agricultural activities also played a certain role. Mn, Cr, Ni, As, Hg, and Cu were mainly affected by natural factors, while Pb, Zn, Cd, and Mo were affected by multiple factors, such as nature and traffic.

**Keywords:** eastern Tibet; surface soil; heavy metal; multivariate statistics; sources; risk assessment





## 1. Introduction

Heavy metals are ubiquitous in the soil environment and are characterized by refractory degradation, long-lasting harm, irreversible enrichment, and food chain enrichment. Through crop enrichment, they reduce crop quality and yield, indirectly affecting human and animal health and endangering the entire ecosystem [1,2]. The sources of heavy metals in soil are diverse, from natural factors, such as rock weathering, to human activities, such as the use of chemical fertilizers and pesticides, mineral exploitation and smelting, transportation, and electroplating [3–6]. Heavy metals such as Pb, Zn, Cd, Cu, and Cr were classified as major controlled pollutants by the US environmental protection agency [7]. The distribution, source, and risk assessment of heavy metals in soil have always been the focus of scientists. Different geological backgrounds, topographies, climates, and anthropogenic activities influence the distribution of heavy metals. [3–6,8–10].

The Tibet Plateau has a unique geographical location, a cold climate, and a high average altitude, with monotonous industrial and agricultural activities, which are less affected by human activities. However, in recent years, with the development of secondary and tertiary industries, the Tibet Plateau has suffered from heavy metal pollution to a certain extent. [11,12] Multiple studies have shown that heavy metal pollution on both sides of the major roads in Tibet has significantly increased [13], and the level of heavy metal pollution in cultivated land has been aggravated by chemical fertilizers and pesticides [11].

At present, the heavy metal soil survey samples in the whole of the Tibet Plateau are distributed near the traffic trunk lines, and the sample points are relatively sparse, which cannot objectively and completely reflect the distribution characteristics in the

region [11,13,14]. Previous studies on eastern Tibet mainly focused on geology, agriculture, ecology, and the water's environment [15–19], and there are few special studies on heavy metal soils in eastern Tibet, a region with active economic activities and a link to the Yunnan and Sichuan provinces.

This study assessed the environmental risk of heavy metals in eastern Tibet by measuring the contents of As, Hg, Pb, Cr, Mn, Mo, Ni, Cu, Zn, and Cd in the surface soil and delineating their main sources using GIS and multivariate statistical methods.

## 2. Materials and Methods

### 2.1. Study Area

The Changdu region is located in the east part of Tibet, which belongs to the Tibet autonomous region of China, located in the Hengduan Mountains and the "three rivers" (the Jinsha river, the Lancang river, and the Nu river) basin area, with typical alpine canyon landforms. The valley is deeply cut into the Hengduan Mountains, showing a typical "V" shape. [20,21] The study area has a continental plateau monsoon climate with thin air. The average annual temperature is 6.5 °C, with a large diurnal temperature difference. The average annual precipitation is 480 mm, and snow is abundant. From high to low altitude, the soil types mainly include cold desert soil, meadow soil, dark brown soil, brown soil, brown soil, and meadow soil, with a low development degree and mainly physical weathering. The study area belongs to the Changdu-Simao stratigraphic area of the Qiangtang-Sanjiang tectonic, stratigraphic region, the main outcrops of the Quaternary, Jurassic, Triassic, Carboniferous, and Palaeozoic-Proterozoic strata (Figure 1 [22]). The main rock types are clastic rocks, limestone, sandstone, shale, mudstone, schist, gneiss, and acidic to silicic intermediate-acidic intrusives. The Dingqing Ophiolite suite is located in the northwest of the study area and belongs to Triassic to Jurassic strata, with ultramafic rocks as the main rock type. Thermal springs and ores, including copper, gold, silver, lead-zinc, mercury, coal, and other minerals, exist in and around the study area. [23–25]

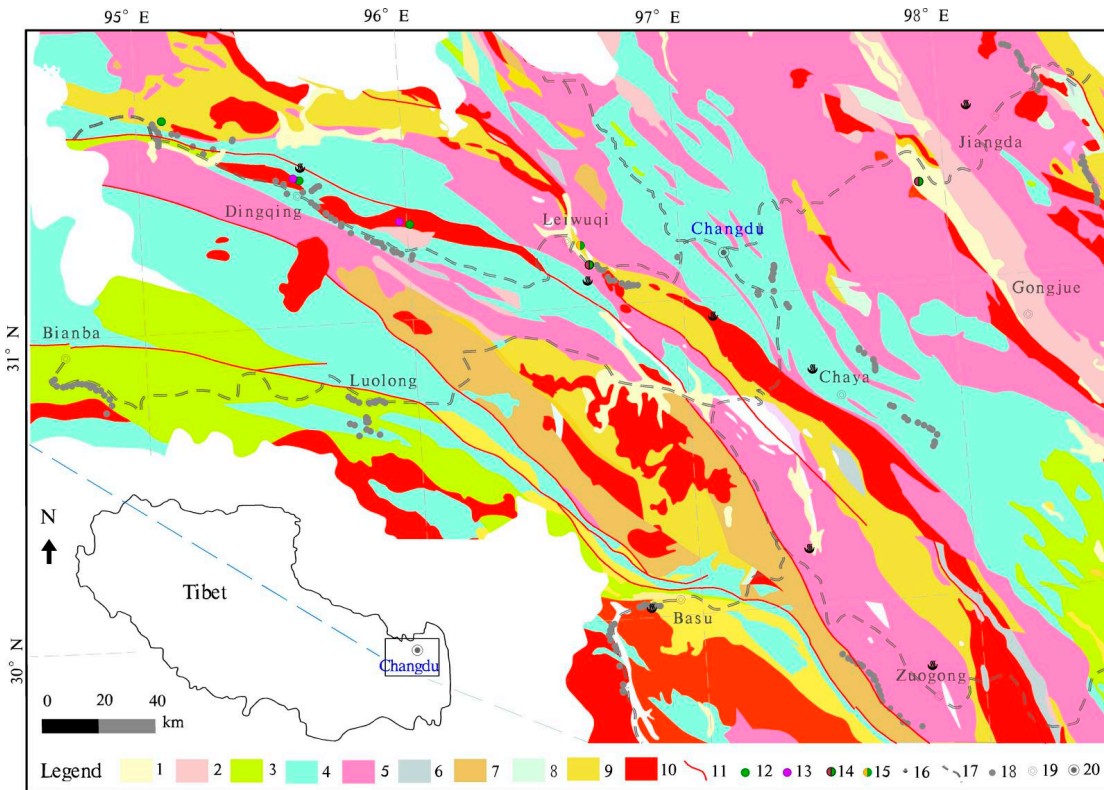

**Figure 1.** Geological mineral map of eastern Tibet and the sampling sites (world geodetic system, 1984 coordinate system). 1. Quaternary sedimentary rocks; 2. Eocene, muddy, sandstone, and gritstone; 3. Cretaceous, clastic rocks, phyllite, and schist; 4. Jurassic, clastic rocks, limestone, sandstone, and shale;

5. Triassic, sandstone, limestone, volcanoclastic rock, slate, coal seam; 6. Permian, coal-bearing clastic rocks, limestone, chert, and subaerial basalt; 7. Carboniferous, limestone, sandy shale, clastic rocks, volcanic, and phyllite; 8. Silurian-Devonian, limestone, shale, sandstone, and intermediate-basic volcanic; 9. Palaeozoic-Proterozoic, schist, gneiss, leptynite, migmatite, and volcanic; 10. Acidic to silicic intermediate-acidic intrusives, granite, diorite, and syenite; 11. fault; 12. chrome ore; 13. nickel ore; 14. copper ore; 15. tin ore; 16. hot spring; 17. road; 18. soil sample point; 19. city; 20. county.

### 2.2. Sampling

In the study area, 205 surface soil samples (with a depth of 0 to 20 cm) of the cultivated land concentrated area were collected from nine counties, including the Karuo district, Siwuqi county, Chaya county, Zuokong county, Badu county, Luolong county, Bian Ba county, Dingqing county, and Jiangda county. The sampling points' distribution is shown in Figure 1.

### 2.3. Laboratory Analysis

After natural air drying in the field processing room, the samples were finely ground with agate mortars and passed through 200 mesh sieves. According to the quartering method, no less than 200 g was taken and sent to the laboratory. The Hebei Geological Testing Center performed the sample test. Mn, Cu, Pb, Zn, Ni, and Cr were measured using X-ray fluorescence spectrometry (XRF, Nihon Co., Ltd., ZSX Primus II, Tokyo, Japan). Mo and Cd were measured using inductively coupled plasma mass spectrometry (ICP-MS, Agilent, 7700x, Santa Clara, CA, USA). Hg and As were measured with atomic fluorescence spectrometry (AFS, Beijing Haiguang Instrument Co., Ltd., 3000, Beijing, China). The test methods and quality control parameters of each trace element are shown in Table 1. The quality of the test methods was controlled by precision, accuracy, and duplicate samples, which met the specification requirements.

**Table 1.** Detection limits and precision of each parameter analysis method.

| Parameter | Analysis Method | Detection Limit (mg/kg) (µg/g) | ΔlgC | RSD (%) |
|---|---|---|---|---|
| Mn | | 2.1 | −0.004~0.003 | 0.88~2.07 |
| Cu | | 1 | −0.020~0.030 | 1.6~8 |
| Pb | XRF | 0.05 | −0.001~0.011 | 1.12~2.84 |
| Zn | | 4 | −0.016~0.006 | 0.5~2 |
| Ni | | 1.78 | 0.000~0.011 | 1.58~2.76 |
| Cr | | 4.44 | −0.009~0.024 | 1~4.7 |
| Mo | ICP-MS | 0.12 | −0.018~0.009 | 1.1~2.96 |
| Cd | | 0.01 | −0.009~0.020 | 1.1~2.96 |
| Hg | AFS | 0.00023 | −0.043~0.037 | 2.5~6.6 |
| As | | 0.0005 | −0.050~0.020 | 1.6~4.1 |

### 2.4. Accumulation Assessment

Which elements in eastern Tibet's topsoils may result in harmful effects? Where are the areas with higher risk? Three indexes, namely the geoaccumulation index (Igeo), enrichment factor (EF), and Nemero synthesis index (PN), were employed to evaluate the possible environmental risks.

EF is a helpful tool for identifying the natural and human sources of metals [26]. To evaluate the enrichment level and anthropogenic influence on soils, the enrichment factor was calculated using Equation (1) [27,28].

$$EF = \frac{(C_i/C_{ref})_{sample}}{(C_i/C_{ref})_{background}} \tag{1}$$

where $C_i/C_{ref}$ is the content ratio of a particular element, $C_i$ is the normalized element, and $C_{ref}$ is the soil sample and the reference background. Mn is usually chosen as the normalized reference element due to its large soil component, low volatility, and immobility [29,30].

Igeo enables the assessment of soil contamination:

$$I_{geo} = log_2(C_i/1.5B_i) \qquad (2)$$

where $C_i$ is the content of a specific element in the topsoil, while $B_i$ is the content of the same elements from Sheng et al. (2012) [11]. The constant 1.5 allows us to analyze the natural fluctuations in the content of a specific substance in the environment [21]. The seven Igeo classes represent the increasing levels of soil contamination (Table 2).

**Table 2.** Enrichment factors (EFs), geoaccumulation (Igeo), Nemero synthesis indices (PN), and contamination grades.

| Enrichment Factor (EF) | Accumulation Degree | Geoaccumulation Index (Igeo) | Pollution Level | Nemero Synthesis Index (PN) | Pollution Level |
|---|---|---|---|---|---|
| EF < 2 | Minor enrichment | Igeo ≤ 0 | Practically none | PN ≤ 0.7 | Pollution-free |
| 2 ≤ EF < 5 | Moderate enrichment | 0 < Igeo ≤ 1 | Unpolluted | 0.7 < PN ≤ 1 | Attention |
| 5 ≤ EF < 20 | Severe enrichment | 1 < Igeo ≤ 2 | Unpolluted to moderately polluted | 1 < PN ≤ 2 | Low pollution |
| 20 ≤ EF < 40 | Very severe enrichment | 2 < Igeo ≤ 3 | Moderately polluted | 2 < PN ≤ 3 | Moderately polluted |
| 40 ≤ EF | Extremely severe enrichment | 3 < Igeo ≤ 4 | Highly polluted | 3 < PN | Highly polluted |
| | | 4 < Igeo ≤ 5 | Highly to very highly polluted | | |
| | | 5 < Igeo | Very highly polluted | | |

To obtain the potential contaminated area, the *PN* method is usually applied [31,32]. The *PN* value for an individual sample was calculated using Equation (3):

$$PN_j = \sqrt{\frac{\left(\frac{C_{ij}}{S_{ij}}\right)^2_{max} + \left(\frac{C_{ij}}{S_{ij}}\right)^2_{ave}}{2}} \qquad (3)$$

where $PN_j$ is the synthesis evaluation score corresponding to the $j$th sample, $S_{ij}$ is the evaluation criterion of the $i$th kind of element for the $j$th sample, and $C_{ij}$ is the measured content value of the $i$th kind of element for the $j$th sample. Each sample site had a *PN* value of its own, and the PN represented the integrated risk loads of all studied elements.

The elements' contents obtained in the late 1970s [14] were taken as criterion values. The levels listed in Table 2 were classified based on the Chinese soil environmental quality assessment standard for green-food production areas [33].

### 2.5. Statistical Analysis

All statistical analyses and plots were carried out using SPSS v 17.0, Origin v 2018, and ArcGIS V 9.2. A statistical program, SPSS v 17.0, evaluated the relationships between various physiochemical parameters and the principal component analysis. Origin v 2018 was used to plot the box-line diagram. ArcGIS V 9.2 was used to determine the spatial distributions of the EF, Igeo, and PN values.

## 3. Results and Discussion

### 3.1. Overview of the Heavy Metal Contents in Eastern Tibetan Soils

The basic statistics for the raw data of eastern Tibet are shown in Table 3. Compared with the upper continental crust (UCC) [34], only the Mo element is low in the topsoil of eastern Tibet, and the contents of As, Pb, Cr, and Ni are significantly higher, which are 12, 2.3, 3.2, and 3.8 times the former respectively; the Hg, Zn, Mn, Cd, and Cu values are significantly higher. Compared with the world soil background value [35], the contents of

Ni, Hg, Cr, As, and other elements are higher, the contents of Cu, Zn, and Pb are similar, and the contents of Cd, Mo, and Mn are low. Compared with the Chinese soil background value [36], the contents of Mo are low, the contents of Cu, Zn, Hg, and Mn are slightly higher, and the contents of Pb, Cr, Cd, As, and Ni are higher. Compared with the soil of the Tibet Plateau in the 1970s [14], the contents of Ni, Cd, and Hg were significantly higher, the contents of As, Pb, Zn, Cu, Mn, and Cr were slightly higher, and the contents of Mo were lower. Compared with the Tibet soil in the early 21st century [11], Cu, Pb, As, Zn, Mn, and Ni were slightly higher, Cd and Hg were significantly higher, and Cr was lower. Ni, Cd, Hg, and other elements were enriched in the soil of the Tibet Plateau, which were significantly different from the UCC, world soil background value, Chinese soil background value, and the soil of the Tibet Plateau in the 1970s. Sheng et al. (2012) [11] provided the latest result of the research on the background value of heavy metals in Tibet. The sample distribution was relatively uniform, and the whole region of Tibet was controlled, so this result was used as the background value in this paper. However, the study did not contain the Mo element, so the content in the 1970s [14] was used as a reference.

**Table 3.** Descriptive statistics of the heavy metals in the soil of eastern Tibet (mg/kg).

| Element | This Study | | | | | | | UCC [c] | Background Values [d] | China [e] | Tibet [f] | Tibet [g] |
|---|---|---|---|---|---|---|---|---|---|---|---|---|
| | Min | Max | Mean | SD [a] | CV [b] | Skewness | Kurtosis | Mean | Mean | Mean | Mean | Mean |
| As | 5.38 | 341 | 24 | 25.3 | 1.054 | 9.87 | 121 | 2 | 7.2 | 11.2 | 18.7 [f] | 19.3 |
| Hg | 0.008 | 0.565 | 0.091 | 82.8 | 0.913 | 3.02 | 11.6 | 0.056 | 0.06 | 0.07 | 0.026 | 0.037 |
| Pb | 9.8 | 262 | 39.3 | 26.9 | 0.683 | 5.32 | 36.7 | 17 | 35 | 26 | 28.9 | 32.2 |
| Zn | 48.5 | 169 | 95.7 | 17.3 | 0.181 | 0.63 | 1.38 | 52 | 90 | 74.2 | 73.7 | 75.6 |
| Cd | 0.08 | 0.68 | 0.2 | 0.08 | 0.389 | 1.99 | 7.34 | 0.102 | 0.35 | 0.1 | 0.08 | 0.141 |
| Cr | 16.5 | 1397 | 111 | 156 | 1.41 | 5.12 | 32.5 | 35 | 70 | 61 | 77.4 | 156 |
| Cu | 8.55 | 67.8 | 28.4 | 8.01 | 0.282 | 1.11 | 3.44 | 14.3 | 30 | 22.6 | 21.9 | 24.3 |
| Ni | 7 | 1457 | 71.4 | 141 | 1.98 | 6.38 | 52.1 | 18.6 | 50 | 26.9 | 32.1 | 55.9 |
| Mn | 466 | 1266 | 783 | 158 | 0.202 | 0.48 | 0.18 | 527 | 1000 | 583 | 626 | 617 |
| Mo | 0.3 | 2.08 | 0.77 | 0.26 | 0.341 | 1.44 | 4.3 | 1.45 | 1.2 | 2 | 1.14 | *ND* |

*ND*: no data. [a] SD: standard deviation; [b] CV: coefficient of variation; [c] Element contents in the upper continental crust [34]; [d] Background values of the world's soils [35]; [e] Element contents in China [36]; [f] Element contents of Tibetan Plateau soils [14]; [g] Element contents of Tibetan Plateau soils [11].

The variation coefficient of heavy metals in surface soil was roughly small, and the order was Ni > Cr > As > Hg > Pb > Cd > Mo > Cu > Mn > Zn. Ni, Cr, and As all showed extreme variations (>100%), while Hg and Pb showed moderate variation (50%–100%) with strong spatial variation. It was preliminarily speculated that the above five elements may be affected by typical rock strata, mineralization, or anthropogenic pollution and are related to the Sanjiang metallogenic belt, structure, or human activities. Additionally, the high geological background should be the main factor. For elements with a higher coefficient of variation, element kurtosis, and skewness, the average content of the elements was significantly higher than the corresponding background value, indicating that human activities may have a greater impact [37,38].

*3.2. Environmental Risk Assessment*

3.2.1. Enrichment Factors

To evaluate which elements showed relatively higher risks in eastern Tibetan soils, the enrichment factor (EF) and geoaccumulation index (Igeo) were calculated.

The order of heavy metal enrichment in the surface soil of eastern Tibet was as follows (Figure 2a): Hg (2.01) > Cd (1.16) > Zn (1.03) > Pb, As, Mn > Ni (0.95) > Cu (0.94) > Mo (0.55) > Cr (0.54). Hg had a moderate enrichment. Cd, Zn, Pb, As, Mn, Ni, Cu, Mo, and Cr were all low enrichment. The moderately enriched points were scattered in the northwest of the study area. The points with higher enrichments of Pb and Cd were caused by traffic and mining factors [11].

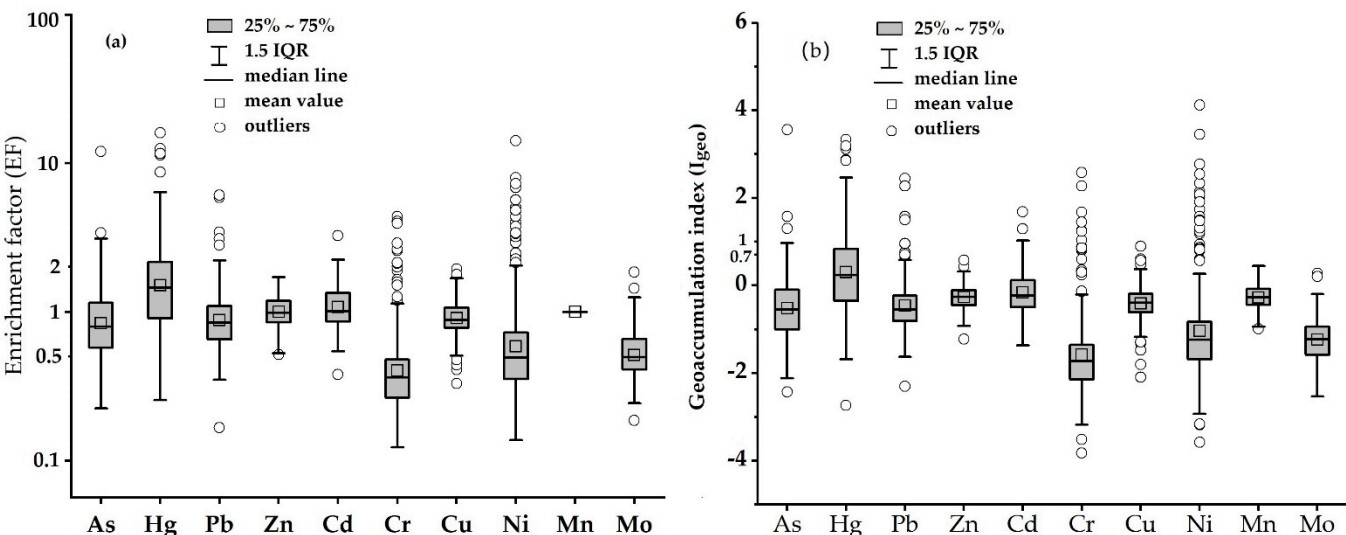

**Figure 2.** Box-line diagram of heavy metal EFs (**a**) and I$_{geo}$ (**b**).

The EF values of Hg and Cd were greater than one, which indicated the existence of the enrichment phenomenon. Furthermore, the EF values of Zn, Pb, As, Mn, Ni, and Cu were close to one, which means the contents of these elements were similar to the background. The relative contents of Mo and Cr were low, with EF values less than one.

3.2.2. Geoaccumulation Index

The geocumulative index analysis results of the study area (Figure 2b) showed that the heavy metals in the soil of eastern Tibet ranged from unpolluted to seriously polluted ($-3.82 <$ Igeo $< 4.12$).

The pollution degree sorting was Hg (0.31) > Cd (0.16) > Zn (0.26) > Mn (0.27) > Cu (0.41) > Pb (0.45) > As (0.52) > Ni (1.04) > Mo (1.23) > Cr (1.57). The heavy metal pollution of the surface soil in eastern Tibet was mainly Hg ($-2.74$–3.33), which was consistent with the EF analysis results. The mercury pollution degree was not high, but the distribution was relatively wide; 17.07%, 3.41%, and 1.46% of the points in the moderate pollution, moderate pollution to serious pollution, and serious pollution, respectively. Ni ($-3.58$~4.12) was not polluted on the whole, but some points had a high pollution degree, 5.37%, 4.88%, 2.93%, 0.49%, and 0.49% are in the state of unpolluted to moderate pollution, moderate pollution, moderate pollution to serious pollution, serious pollution, and extreme pollution respectively, with relatively concentrated distributions. They were mainly located in the northwest of the study area. Cd ($-1.37$–1.68) was in the overall unpolluted state, with 32.20% and 1.95% in the unpolluted to moderately polluted state, and moderately polluted state, respectively. The pollution degree was low, but the distribution was relatively scattered. More than 80% of the other seven elements were uncontaminated. Parts of the contaminated sample points are shown in Figure 3. Some of the points in the central and northern parts showed a combined pollution phenomenon.

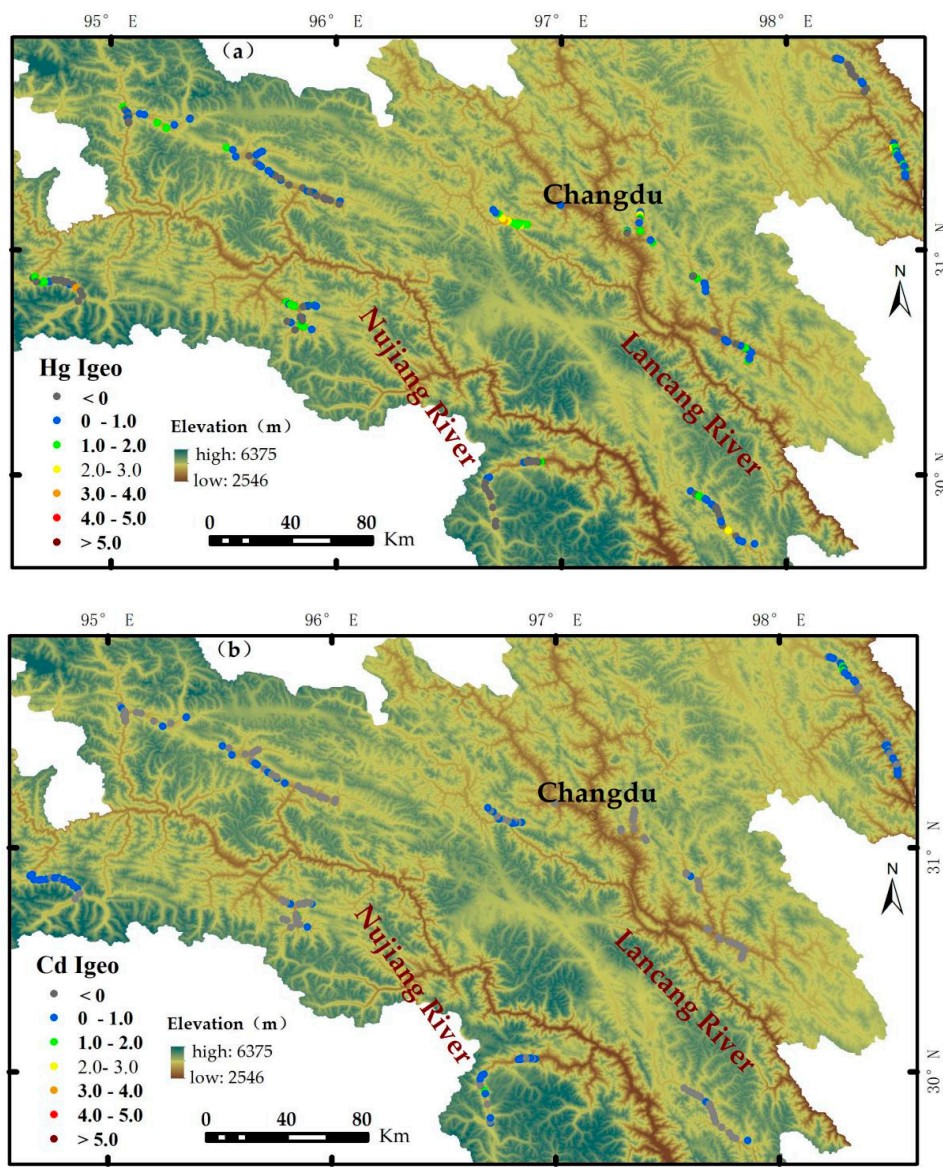

**Figure 3.** Spatial distribution of geoaccumulation of Hg (**a**) and Cd (**b**).

### 3.2.3. Nemero Synthesis Index (PN)

The PN results showed that 1.98% of the surface soil samples in eastern Tibet were at the alert level (Figure 4), and the proportions of the low pollution, medium pollution, and serious pollution levels were 52.68%, 20.98%, and 24.39%, respectively. The overall level of this comprehensive evaluation was comparable to Sheng et al. (2012) [11]. Eastern Tibet belongs to an area with a high geological background of heavy metals, in which the pollution phenomenon is widespread, and the environmental conditions vary from low pollution to severe pollution. Nevertheless, the results of the enrichment factors (EF) and geoaccumulation indexes (Igeo) showed that there was only sporadic low-level pollution. After the comparison and analysis with the PN, it was found that the three analysis results were consistent in the polluted area, but the pollution degree of the PN was higher, especially in the northwest, which was related to the consideration of the average and maximum contents of elements in the PN calculation [13,39]. The high PN values of this area were dominated by contents of Hg, Cd, and Ni, which occupied 51.22%, 14.15%, and 12.20% of the maximum terms in Formula (3), respectively. The higher pollution levels were because the maximum value was considered.

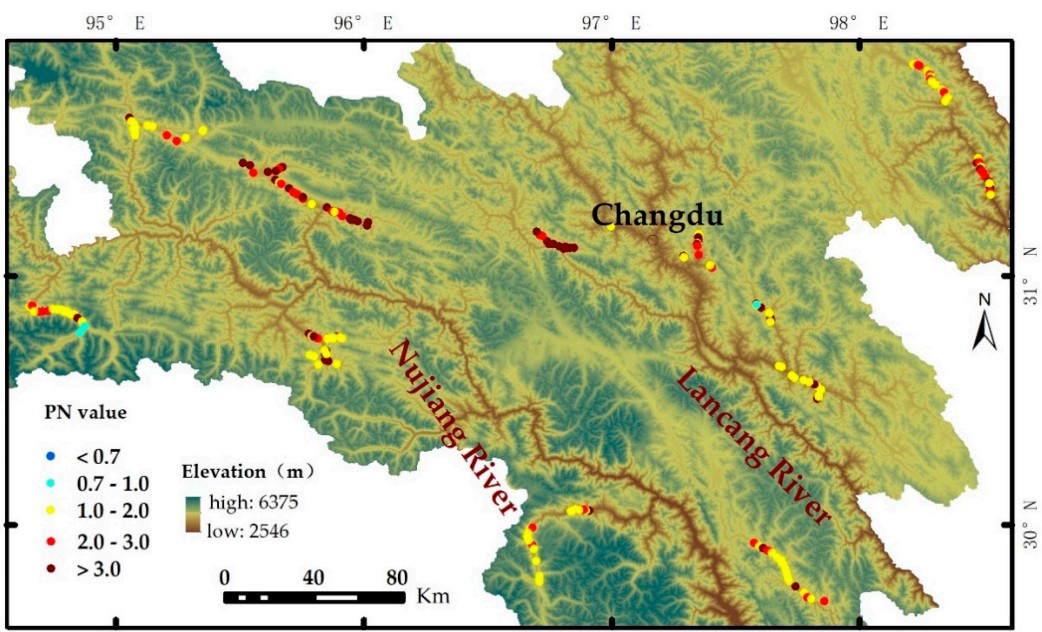

**Figure 4.** Interpolated PN values of the Nemero synthesis index.

Based on the analysis results of the above pollution indexes, heavy metal pollution was sporadic in eastern Tibet and relatively concentrated in the central and northern parts. The pollution may be the result of rock weathering, mining, and agricultural activities. At the same time, the sampling sites were mainly located near traffic trunk lines, which were jointly affected by high traffic volume, high emissions, and low oxygen high accumulation [11,37].

### 3.3. Multivariate Statistical Analysis

#### 3.3.1. Correlation Analysis

Elemental correlation analysis is one of the important bases for predicting the origin of heavy metals in soil, and a significant correlation between heavy metals in soil indicates the existence of the same or similar source [40]. As shown in Table 4, the pairwise correlation of the three groups (As, Pb, Zn, Cd, and Mo; Hg and Cu; Mn, Cr, and Ni) were highly significant ($p < 0.01$). Because the ten elements were not completely and significantly correlated, there may be multiple sources. Mn, Cr, and Ni may be related to the weathering of ultramafic rocks and chrome ore mining [12]. As, Pb, Zn, Cd, and Mo may be affected by factors such as parent material and transportation [41]. Hg and Cu may be related to agricultural activities and the exploitation of hot springs and copper deposits.

**Table 4.** Correlation coefficients among the heavy metals.

|  | **As** | **Hg** | **Pb** | **Zn** | **Cd** | **Cr** | **Cu** | **Ni** | **Mn** | **Mo** |
|---|---|---|---|---|---|---|---|---|---|---|
| As | 1 | | | | | | | | | |
| Hg | 0.104 | 1 | | | | | | | | |
| Pb | 0.195 ** | 0.187 ** | 1 | | | | | | | |
| Zn | 0.422 ** | 0.121 | 0.356 ** | 1 | | | | | | |
| Cd | 0.562 ** | 0.047 | 0.293 ** | 0.543 ** | 1 | | | | | |
| Cr | −0.111 | −0.107 | −0.197 ** | −0.126 | −0.202 ** | 1 | | | | |
| Cu | 0.084 | 0.213 ** | 0.245 ** | 0.503 ** | 0.329 ** | −0.115 | 1 | | | |
| Ni | −0.108 | −0.109 | −0.192 ** | −0.171 * | −0.196 ** | 0.928 ** | −0.162 * | 1 | | |
| Mn | 0.081 | −0.01 | 0.001 | 0.371 ** | 0.211 ** | 0.306 ** | 0.363 ** | 0.283 ** | 1 | |
| Mo | 0.255 ** | −0.096 | 0.093 | 0.1 | 0.412 ** | −0.159 * | −0.098 | −0.118 | −0.017 | 1 |

** Significant at the 0.01 level. * Significant at the 0.05 level.

#### 3.3.2. Principal Component Analysis (PCA)

Bartlett's test of sphericity and the KMO test was used to test whether the elements were suitable for the principal component analysis. The results of the Bartlett sphericity test (significance 0.00 < 0.05) and KMO measurement test (0.657 > 0.5) of the heavy metal

elements in the surface soil of east Tibet showed that the principal component analysis was suitable for each element [42–44]. After the orthogonal rotation of Kkaise normalized factors by Varimax, four principal components with eigenvalues greater than one were obtained, and their contribution rates reached 74.31%, which could reflect the main information of all heavy metals, while the contribution rates of the other components were less than 8%.

The PCA is useful to further explore the potential sources of heavy metals in soils. Four PCs were extracted from the rotated component matrix for the PCA (Table 5), accounting for 74.31 % of the total variance. Based on the PC loadings, ten elements could be grouped into four PCs (F1–F4). PC1 (Cu, Cd, Pb, and Zn), PC2 (Cr, Ni, and Mn), PC3 (Mo), and PC4 (As and Hg), which accounted for 33.24%, 19.20%, 11.55%, and 10.32% of the total, respectively.

**Table 5.** Principal component analysis matrix.

| Element | F1 | F2 | F3 | F4 |
|---|---|---|---|---|
| As | 0.35 | 0.326 | 0.04 | 0.718 |
| Hg | 0.197 | 0.084 | −0.719 | 0.435 |
| Pb | 0.548 | 0.132 | −0.18 | −0.054 |
| Zn | 0.643 | 0.579 | −0.041 | −0.108 |
| Cd | 0.709 | 0.359 | 0.32 | 0.097 |
| Cr | −0.799 | 0.51 | 0.077 | 0.102 |
| Cu | 0.589 | 0.374 | −0.245 | −0.459 |
| Ni | −0.813 | 0.485 | 0.087 | 0.113 |
| Mn | −0.246 | 0.825 | 0.062 | −0.186 |
| Mo | 0.489 | −0.092 | 0.65 | 0.185 |
| Initial eigenvalue | 3.324 | 1.920 | 1.155 | 1.032 |
| Variance contribution rate, % | 33.24 | 19.20 | 11.55 | 10.32 |

The contribution rate of the first principal component (F1) was higher than that of the other main components. The heavy metal elements with high loads were Pb, Zn, Cd, and Cu, which were 0.548, 0.643, 0.709, and 0.589, respectively. Except for lead, the coefficients of variation of the other three elements were low (Table 3), indicating that anthropogenic activities had little influence. Pb, Zn, Cd, and Cu were significantly correlated, suggesting that these four elements had the same or similar source. Pb, Zn, Cd, and Cu were greatly influenced by the soil parent material in Tibet [34]. Meanwhile, studies have shown that the Pb, Zn, and Cd contents are high on the surface along the highways in Tibet, and the incomplete combustion of gasoline and the release of tire wear and tear [11,13,39] indicate that there may be two sources of natural factors and traffic factors.

The contribution rate of the second principal component (F2) was 19.20%. The heavy metal elements with higher loads were Cr, Ni, and Mn, which were 0.51, 0.485, and 0.825, respectively. Mn, Cr and Ni showed a significant correlation ($p < 0.01$), indicating a common source. Mn is abundant in the Earth's crust, and the soil of ultramafic rocks in the Qinghai-Tibet Plateau had a high Cr and Ni background [12], and the study area had chrome ore, nickel ore, and ophiolite suite, suggesting that the composition of this group is mainly affected by natural factors.

The heavy metal element with a higher load of the third principal component (F3) was Mo (0.65). Mo was mainly affected by soil parent material and traffic pollution [45], mining, and metallurgy [46]. Mo was significantly correlated with As and Cd (Table 4). Moreover, there are many non-ferrous metal deposits in eastern Tibet [22], so F3 may have been affected by comprehensive factors.

The heavy metal elements with a high load of the fourth principal component (F4) included As and Hg, which were 0.718 and 0.415, respectively. Due to their association with widely distributed shales and hot springs in the study area, they provide material sources of arsenic and mercury [47].

## 4. Conclusions

The contents of mercury and nine metals, including Mo, Cr, Mn, Ni, As, Cu, Zn, Cd, and Pb, were obtained from 205 surface soils in east Tibet, China. Moreover, the heavy metal composition and main ion-controlling factors were analyzed. The main conclusions that could be drawn are as follows:

(1) Compared with the upper continental crust (UCC) background, the world soil background, the Chinese soil background, and the Tibetan Plateau soil in the 1970s, As, Ni, Cd, Hg, and other elements were enriched to different degrees in the eastern Tibetan Plateau soil, while Mo elements were depleted;

(2) Compared with the contents of Qinghai-Tibetan soil in the early 21st century, the contents of Cr in this study were lower, the contents of Cu, Pb, As, Zn, Mn, and Ni were slightly higher, and the contents of Cd and Hg were significantly higher. Ni, Cr, As, Hg, Pb, and other elements had a large coefficient of variation and strong spatial variation, which may be due to the existence of point source pollution;

(3) The analysis of the enrichment factors (EFs), the geoaccumulation index (Igeo), and the Nemero composite index (PN) showed that the soil in eastern Tibet was mainly polluted by Cd and Hg, but the pollution degree was generally not high and the distribution was relatively scattered. The central and northern parts were relatively concentrated, and higher traffic flow and mining activity may be the main causes of pollution. Overall, the soil in the study area was unpolluted. To better criticize the environmental risks, the bioavailability of heavy metals in soil should be measured, and the grassland and forest areas of the plateau should be further considered;

(4) According to principal component analysis, the influencing factors of heavy metals in the soil of eastern Tibet include natural factors, traffic factors, agricultural activity factors, and mining factors, and the former two factors are the main factors. The elements mainly affected by natural factors included Mn, Cr, Ni, As, Hg, Cu, and the elements affected by nature and traffic included Pb, Zn, Cd, and Mo.

**Author Contributions:** Conceptualization, M.W. and L.Y.; methodology, M.W.; software, M.W.; validation, L.Y.; formal analysis, J.L.; investigation, J.L.; resources, J.L.; data curation, J.L.; writing—original draft preparation, M.W.; writing—review and editing, L.Y.; visualization, M.W.; supervision, Q.L.; project administration, M.W. All authors have read and agreed to the published version of the manuscript.

**Funding:** This research was funded by the Geological survey project (DD20190534 and DD20221754) of the China Geological Survey.

**Conflicts of Interest:** The authors declare no conflict of interest.

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
