# Peer review of "The Evaluation and Sources of Heavy Metal Anomalies in the Surface Soil of Eastern Tibet"

_minerals, doi:10.3390/min13010086_

Round 1

Reviewer 1 Report

The paper is scientifically very interesting.

The subject of the study is relevant to the scope of the special issue, i.e. “soil contamination”.

The structure of the paper is excellent.

It has high-quality data.

The authors answer the questions posed in the article.

I would therefore suggest to accept the manuscript for publication after minor revision.

My suggestions to the authors are

1) Sampling paragraph - Your samples were collected in ca. 11 areas. Please explain for which reasons you choose each area.

2) Laboratory analysis paragraph - You ought to mention

i) in which lab you performed the analyses and

ii) the models of the instruments you used.

3) Correlation analysis paragraph -You identified 3 groups; you ought to try to explain these correlations, i.e. relate them with a specific source (specific ore deposit? specific rock formations? etc.).

4) Correct the format of the references, for example, 21, 25

5) Figure 1- include topographic information.

6) Legend of Figure 1- include

i) the used coordinating system and

ii) references based on the geological map that you used. If it is a new mapping, you ought to mention it in the manuscript.

7) Table 3 – 2nd line, correct the “Tibet”

8) Table 5 – correct the position of the table

9) Bibliography - From the international bibliography, please use papers in which the authors are from various countries. (Please see the references below and the references within.)

Kanellopoulos, C. and Argyraki, A., 2013. Soil baseline geochemistry and plant response in areas of complex geology. Application to NW Euboea, Greece. Chemie der Erde – Geochemistry, 73(4), 519-532, https://doi.org/10.1016/j.chemer.2013.06.006

Kanellopoulos, C., Argyraki, A., Mitropoulos, P., 2015. Geochemistry of serpentine agricultural soil and associated groundwater chemistry and vegetation in the area of Atalanti, Greece. Journal of Geochemical Exploration, 158, 22-33, https://doi.org/10.1016/j.gexplo.2015.06.013

Kanellopoulos, C., 2020. Influence of ultramafic rocks and hot springs with travertine depositions on geochemical composition and baseline of soils. Application to eastern central Greece. Geoderma, https://doi.org/10.1016/j.geoderma.2020.114649

Kanellopoulos, C., Argyraki, A., 2022. Multivariate statistical assessment of groundwater in cases with ultramafic rocks and anthropogenic activities influence. Applied Geochemistry 141, 105292, https://doi.org/10.1016/j.apgeochem.2022.105292

10) Also, my suggestion is to include an Annex with your data, i.e. for each sample present

i) coordinates and if you have the soil type

ii) geochemical analyses,

iii)  Enrichment factor(EF), geoaccumulation(Igeo), Nemero synthesis indices (PN), and contamination grades.

11) Please correct minor linguistic errors, read the manuscript carefully.

12) Some texts are in red color

Author Response

Dear reviewer,

We gratefully appreciate your valuable comment, which has significantly raised the manuscript’s quality and enabled us to improve the manuscript. Each comment brought forward by you was accurately incorporated and considered. The comments have been responded point by point, and revisions are indicated.

Comment 1: Sampling paragraph - Your samples were collected in ca. 11 areas. Please explain for which reasons you choose each area.

Reply: In the study area, soil mainly distributed in the valleys, and we selected cultivated land concentrated areas for the sampling. (line 87)

Comment 2: Laboratory analysis paragraph - You ought to mention

  1. i) in which lab you performed the analyses and
  2. ii) the models of the instruments you used.

Reply: The lab (Hebei province geological experimental testing center) has been added in the manuscript (line 136-137), and the model information of the test instruments also. (line 138-141)

Comment 3: Correlation analysis paragraph -You identified 3 groups; you ought to try to explain these correlations, i.e. relate them with a specific source (specific ore deposit? specific rock formations? etc.).

Reply: Done. Preliminary judgment has been made.

“Mn, Cr, and Ni may be related to weathering of ultramafic rocks and chrome ore mining [12]. As, Pb, Zn, Cd and Mo may be affected by factors such as parent material and transportation [41]. Hg and Cu may be related to agricultural activities and the exploitation of hot spring and copper deposits.” (line 467-470)

Comment 4: Correct the format of the references, for example, 21, 25

Reply: Done.

Comment 5: Figure 1- include topographic information.

Reply: The elevation information of the research area has been included in Fig.3 and Fig.4. In order to improve the simplicity of the Fig.1 does not add terrain lines.

Comment 6: Legend of Figure 1- include

  1. i) the used coordinating system and
  2. ii) references based on the geological map that you used. If it is a new mapping, you ought to mention it in the manuscript.

Reply: Done, the coordinating system - wgs84 was used in figure 1, and the information has been added. (line 123-124) And the reference of the geological map has been added. (line 80)

Comment 7: Table 3 – 2nd line, correct the “Tibet”

Reply: Done.

Comment 8: Table 5 – correct the position of the table

Reply: Done.

Comment 9: more Bibliography

Reply: Thank you for your valuable suggestion. We have cited them and more related literature in the revised manuscript's proper place.

Comment 10: Also, my suggestion is to include an Annex with your data

Reply: Sorry, the data are stored in the Geological Archive of Tibet Autonomous Region. Researchers can apply for the data by themselves, but individuals are not allowed to disclose the original data according to the agreement.

Comment 11: Please correct minor linguistic errors, read the manuscript carefully.

Reply: we read the manuscript carefully, and correct several errors, thank you.

Comment 12: Some texts are in red color

Reply: Done.

Yours sincerely

Reviewer 2 Report

Some additional, specific comments:

1. What is the main question addressed by the research?

Determination of heavy metal concentrations in surface soils and their sources in eastern Tibet. However, authors should add information about why they are investigating this problem (who needs this information and for what purposes)?

2. Do you consider the topic original or relevant in the field? Does it address a specific gap in the field?

The topic is relevant in this area for the whole world. This study significantly expands the possibilities of solving the scientific problem under study.

3. What does it add to the subject area compared with other published material?

Sources of influence on the concentration of heavy metals in surface soils were found, including natural factors and industry.

4. What specific improvements should the authors consider regarding the methodology? What further controls should be considered?

At the beginning of the Results section, a detailed comparison of the obtained concentrations of heavy metals with the world background and with the Chinese one is given. However, there is a lot of information and it is desirable to add a graphic representation of this description for greater clarity.

5. Are the conclusions consistent with the evidence and arguments presented and do they address the main question posed?

Conclusions are logical and well founded

6. Please include any additional comments on the tables and figures.

The quality of tables and figures is appropriate

Author Response

Dear reviewer,

We gratefully appreciate your valuable review.

sincerely

Reviewer 3 Report

Hello; your study is extremely interested especially in the field of studies of lord metals in soils and ells presents many especially important results. Just one question, is it not possible to add the other studies that have already been done on the same region? good job and good luck

Author Response

Dear Reviewer,

We gratefully appreciate your valuable comment,  studies that have already been done on the same region have been added in the manuscript. (line 56-58)

Yours sincerely